# A Model for Cognitive Personalization of Microtask Design

**DOI:** 10.3390/s23073571

**Published:** 2023-03-29

**Authors:** Dennis Paulino, Diogo Guimarães, António Correia, José Ribeiro, João Barroso, Hugo Paredes

**Affiliations:** 1School of Science and Technology, University of Trás-os-Montes e Alto Douro, 5000-801 Vila Real, Portugal; dpaulino@utad.pt (D.P.);; 2INESC TEC, 4200-465 Porto, Portugal

**Keywords:** crowdsourcing, cognitive abilities, human-computer interaction, microtask design, personalization, task fingerprinting

## Abstract

The study of data quality in crowdsourcing campaigns is currently a prominent research topic, given the diverse range of participants involved. A potential solution to enhancing data quality processes in crowdsourcing is cognitive personalization, which involves appropriately adapting or assigning tasks based on a crowd worker’s cognitive profile. There are two common methods for assessing a crowd worker’s cognitive profile: administering online cognitive tests, and inferring behavior from task fingerprinting based on user interaction log events. This article presents the findings of a study that investigated the complementarity of both approaches in a microtask scenario, focusing on personalizing task design. The study involved 134 unique crowd workers recruited from a crowdsourcing marketplace. The main objective was to examine how the administration of cognitive ability tests can be used to allocate crowd workers to microtasks with varying levels of difficulty, including the development of a deep learning model. Another goal was to investigate if task fingerprinting can be used to allocate crowd workers to different microtasks in a personalized manner. The results indicated that both objectives were accomplished, validating the usage of cognitive tests and task fingerprinting as effective mechanisms for microtask personalization, including the development of a deep learning model with 95% accuracy in predicting the accuracy of the microtasks. While we achieved an accuracy of 95%, it is important to note that the small dataset size may have limited the model’s performance.

## 1. Introduction

The international classification of functioning disability and health (ICF) encompasses the classification of health information [1]. Studies have been conducted to improve this information, regarding the transparency and reliability of the process of linking health information to the ICF [2]. In recent years, the ICF has been employed to categorize cognition-related information, including cognitive-communication disorders, which entail several challenges in terms of terminology, assessment, and sociocultural context. In this regard, the usage of ICF can lead to significant therapeutic interventions for individuals suffering with this kind of disorder [3]. Gauthier and colleagues [4] described mild cognitive impairment (MCI) and provided a conceptual background on this impairment, the pathophysiology, the tools typically used for the diagnosis, some procedures and statistics regarding the management per patient, and what can be used to prevent it. Dementia in the stage of serious decline can be considered as a mild cognitive impairment, and the worldwide costs are enormous, unevenly distributed, and increasing. Since 2010, dementia costs have reached USD 818 billion globally, an increase of 35% [5]. Technology benefits people with MCI, by providing a means of support [6]. Nowadays, technology can be used to support cognitive rehabilitation by maintaining or even improving an individual’s mental state. Braley and colleagues [7] conducted a study to evaluate the feasibility of constructing smart home systems to help people with dementia in quotidian activities through the use of auto-prompts. The feasibility was validated, and factors such as positive reinforcement, training, and research related to human interaction were identified as necessary for developing these types of systems. Another article proposed a model for assistive technology, joining physical and cognitive rehabilitation [8]. This approach is interesting in merging both approaches and prescribing rehabilitation, giving therapists opportunities to customize the best rehabilitation exercise for their patients.

Furthermore, this approach also comprises a social feature, including collaborative exercises. Another fact that must be taken into consideration regarding technology that supports cognitive rehabilitation is the level of dependency of the user. In a systematic review based on technology-based cognitive rehabilitation for individuals with cognitive impairment, most of the identified studies featured the direct help of a therapist for the participant [9]. The direct help could bias the result or also mean that a design in the form of personal help is essential when developing cognition-aware technology [10]. However, if technology is neither well designed nor cognition-aware, in the case of individuals with MCI, they can commit errors in terms of accuracy and miss the default time windows in tasks, with the latter being problematic, for example with cash machines [11]. Drawing together the findings from prior literature, numerous scholars have argued that the proper design of technology must take into consideration the user’s cognitive abilities.

Cognitive abilities are wider than specific abilities and belong to the group of general mental ability (GMA). As a construct, GMA obtains significant correlation outcomes with occupation level and performance in job training. Even job historical performance has a weaker correlation when compared with GMA. With this framing, workers with a higher GMA acquire more and faster job knowledge, which translates to higher levels of job performance [12]. Cognitive abilities, which include but are not restricted to working memory (managing and storing information at the same time [13]) and executive functions (cognitive processes involved in behavior towards goal accomplishment [14]), can predict performance in most jobs and situations. In Web-based unsupervised environments, the potential for faking non-cognitive and cognitive ability measures underlines the need for caution and there is an ongoing discussion about their harmful effects [15]. In [16], the authors provided an updated taxonomy for cognitive abilities and personality domains. Cognitive abilities can be measured remotely, with potential for the administration of self-reported tools for assessing capabilities in cases where there is a significant personality bias [17]. With this in mind, the utility of the tool is questioned. This reinforces the usage of online tasks (e.g., microtasks in the context of crowdsourcing), including cognitive tests to assess cognitive abilities, instead of using self-measurement tools that are biased by nature. Moreover, IQ can be used to predict a worker’s job satisfaction, as well as expected job performance, while personality type can moderate the previous relationship [18]. It is also worth noting that cognitive abilities can be improved when emotional intelligence is promoted [19]. In this sense, besides cognitive abilities, social-cognitive factors contribute to remarkable differences in technology task performance. In the elderly, computer proficiency can be predicted from scores in cognitive ability tests. Specifically, predictions can be made using sense of control, psychomotor speed, and inductive reasoning [20]. Among the most promising techniques enabling a more personalized experience with online digital labor platforms, this work opens a new perspective for using crowdsourcing in assessing the cognitive abilities of each worker based on cognitive tests, with the ultimate goal of matching tasks and crowd workers’ individual capabilities.

Over the years, crowdsourcing has evolved, both from a technological point of view and regarding research interest, with several literature reviews and surveys having been published in the last decade (e.g., [21,22]). Since the term’s inception in the mid-2000s [23], crowdsourcing has become increasingly prevalent in local and remote settings, where there is a need to obtain timely information for solving simple to moderately complex problems of varying nature and length. In fact, crowd-powered systems have gradually matured across the world, and we can see examples of renowned pioneer companies (e.g., [23]) adopting crowdsourcing as part of their innovation strategy and business model. Considered an important form of on-demand digital labor, crowdsourcing tasks require fewer skills and less time to complete, giving a flexible job opportunity. In line with this, the evolution of digital work has created conditions where people can overcome social and geographical barriers in an inclusive setting, especially in the context of microtask crowdsourcing, due to the ease of performing decomposed tasks. As a result, more and more opportunities arise for supporting people with cognitive or physical impairments to perform remote work [24]. Considering microtasks, people can perform tasks with different levels of complexity via Web or mobile applications, by doing something they are interested in (e.g., play games, transcribe language, or label images) [25].

Task fingerprinting is a technique for identifying behavior traces from crowd worker activity, to improve quality control techniques. The pioneer work conducted by Rzeszotarski and Kittur [26] inferred behavior patterns in a crowd work context by analyzing the user interaction log events, such as the click details and key presses. This method of microtask fingerprinting develops prediction models based on machine learning (ML) to identify the behavioral traits of workers. Aligned with this goal, various research works have been conducted with the aim of developing this line of inquiry. For instance, a study on quality control mechanisms proposed a set of indicators and a general framework covering more types of microtask, including open-ended answers [27]. This work obtained better outcomes when compared to the state-of-the-art methods, such as the traditional analysis of historical performance in crowd work [28]. Furthermore, a supervised ML model was proposed, where more types of crowd worker profile were detected, with a higher granularity [29]. Additionally, a model was created to define behavior, motivation, and performance from a general perspective. Fine-grained features were also analyzed in another study, with the results corroborating their underlying benefits for quality control mechanisms [30].

In addition to traditional quality control mechanisms and task fingerprinting, cognitive personalization can be applied to microtask assignment arrangements. Goncalves and colleagues [31] proposed a method for performing task routing, based on cognitive ability tests. However, while the former was performed on a computer, the latter was administered in pencil and paper. The positive results obtained in this study support the assessment of cognitive ability for routing microtasks. In the following related work, Hettiachchi and colleagues [32] transformed pencil-and-paper cognitive ability tests into microtasks suitable for short crowdsourcing scenarios. However, the cognition-based task routing was not performed in real-time. A thorough study was subsequently performed involving 574 crowd workers, and this time involving real-time task assignment. Predominantly, this research concluded that short-length cognitive tasks supported better outcomes in the task routing when compared to the conventional methods, including validated state-of-the-art microtask assignment methods [33]. While these studies based on cognition-based task assignment obtained excellent results, two pertinent questions arise: can cognitive tests support personalization in the design of microtasks? Task assignment is important in providing microtasks suitable for each crowd worker, but can any microtask be adapted such that each crowd worker is sufficiently motivated to perform it? The answer to these questions will allow an increase in the democracy in crowdsourcing settings, by improving the microtask itself and taking into consideration the adaptation requirements of the crowd worker. First, it is necessary to clarify the roles of personalization and customization in technology. Personalization in computer systems is defined as the process of adjusting the functionality or the interface to increase personal relevance from the user’s perspective. Customization refers to providing customization options for the user to adjust the technology in accordance with his/her preferences. A significant difference between personalization and customization is that personalization is done implicitly by the system, while customization is done explicitly by the user. Although personalization reduces the user burden in the personal adaptation of the system, it can cause privacy-related problems. One solution to tackle these privacy issues and reduce the user burden is to elicit user adaptation preferences and requirements through small interactive tasks [34]. This can be applied and adaptated in the design of microtasks, through cognitive tests being transformed into microtasks and complemented with task fingerprinting.

For this study, we recruited 134 unique crowd workers in four different conditions (below median cognitive score, above or equal median cognitive score, microtasks without personalization, microtasks with personalization), in order to answer the following research questions (RQs):RQ1: How can the administration of cognitive ability tests, specifically in the evaluation of executive functions, be used to allocate crowd workers to microtasks with or without personalization?RQ1.1: Is a deep learning model able to allocate crowd workers to microtasks with different characteristics of UI and complexities of the content presented?RQ2: Can task fingerprinting, based on the identification of behavioral traces, be used to allocate crowd workers to microtasks with or without personalization?

To answer these questions, we deployed a set of batches comprised of cognitive tests (based on cognitive abilities, specifically executive functions) and prototypical microtasks in two human intelligence tasks (HITs). Following a between-subjects study design, the HITs differed in meeting two conditions: microtasks without personalization (normal difficulty) and with personalization (easy difficulty, where the inputs and/or the content was simplified). Both HITs had the same cognitive tests and only differed in terms of difficulty. The main purpose of this study was to investigate whether it is feasible to allocate crowd workers to suitable microtasks based on cognitive tests and task fingerprinting. A secondary purpose was to investigate task fingerprinting, not only to obtain behavioral traces of crowd workers from a performance standpoint, but also to complement the evaluation made through the administration of cognitive ability tests.

## 2. Background

The ground-breaking works of Hettiachchi and colleagues [32,33] revealed the underlying potential of cognition-based microtask assignment. However, in addition to microtask assignment, there have been other approaches to performing cognitive personalization of microtask design. Eickhoff [35] conducted a study to identify the cognitive biases (i.e., systematic errors in the thinking processes) of crowd workers and indicated that microtask design should take these biases into consideration, in order to avoid a decrease in terms of work performance. In another work based on collaboration scenarios, a model was developed using the random forest algorithm to identify relevant collaboration skills [36]. Moreover, Sampath et al. [37] found that performance in text-transcription tasks can be improved significantly if the microtask design takes into account working memory or visual saliency. Paulino and colleagues [38,39] suggested that cognitive styles could be used to infer information processing preferences in a crowd work setting. These preferences can then be used to personalize the microtask interface. However, these studies can be considered analyses of user log interactions to enhance cognitive personalization.

Besides the seminal work of Rzeszotarski and Kittur [26], other similar techniques have been developed for identifying behavioral traces of crowd workers. A system was proposed to integrate different techniques for quality control in the context of crowdsourcing [40]. This system allowed the integration of outcomes related to gold standards, majority voting, and behavioral traces, with the generation of graphics and other forms of data visualization. Another study based on behavioral data captured from logging mouse interactions and eye tracking data showed that this is beneficial and can complement task duration metric for quality control purposes [41]. Additional behavioral traces from crowd workers can be identified, to develop a model for predicting label effectiveness, as well as worker performance. A previous work more focused on classification microtasks generated a model for optimizing label aggregation processes [42]. Furthermore, it was found that the classification accuracy can also be improved significantly when using gold judges based on behavioral data [43]. Regarding the data collected, most of the studies identified behavioral traces based on the raw data from clicks or key presses. In the pioneering work of Rzeszotarski and Kittur [26], the features identified were the simple interactions made in the microtask interface, such as the page scrolls, the special keys presses (e.g., tab or enter), and the change of focus, either on the input fields or in the browser tabs. To complement these indicators, the timestamp to the millisecond was identified for each interaction. This basic approach to processing raw data was used in other studies [27,42]. Additionally, Han and associates [27] created a browser extension to be used by crowd workers when performing microtasks. The authors extended the temporal and behavior indicators from the work of Rzeszotarski and Kittur [26] but provided new features regarding contextual and compound features. The contextual features refer to the study of two or more behaviors simultaneously, while the compound features are related to the analysis of a sequence of interactions that enrich behavioral identification (e.g., if a crowd worker scrolls the page frequently, it may indicate that he/she is hesitant). In another study, Gadiraju and colleagues [29] complemented behavioral trace identification by analyzing the time before, during, and after interactions with finer granularity.

While there has been significant progress in task fingerprinting in crowd work, especially by Rzeszotarski and Kittur [26] and later by Gadiraju and co-authors [29], a research gap exists in combining crowd workers’ interaction logs with cognitive tests. This combination has the potential to enhance the performance of crowd workers and improve the quality of work delivered to the requesters.

## 3. Methodology

A case study was designed to answer the previously formulated research questions. In this section, the methodology of this case study will be presented. There are two ways to achieve cognitive personalization in microtask design: administering cognitive tests to assess capabilities, or using task fingerprinting. The former involves transforming psychometric tests into microtasks, a method successfully used in previous studies [32,33,38,39]. The latter involves evaluating the behavior of crowd workers while performing microtasks, which can be done by analyzing click details or keyboard press data [26,44]. This study used both methods: psychometric tests and microtask fingerprinting. Combining these methods can be mutually beneficial, since behavioral traces identified in microtask fingerprinting can enhance the accuracy of psychometric tests, and the results from psychometric tests can support the identification of new behavioral traces.

This case study was established using a prototypical microtask crowdsourcing scenario, based on methods and subsequent findings described in [33]. First, the tasks that were presented to the crowd workers were based on cognitive tests (see Table 1 for a detailed outline) and evaluated executive functions that have been proven to be effective in assessing the cognitive abilities of crowd workers [33]. These cognitive tests were transformed into short microtasks, each one made up of several small trials in the form of cognitive puzzles, which were appropriate to a crowd work context. Second, the prototypical microtasks were also based on the work of Hettiachchi et al. [32,33], who acquired positive outcomes concerning the combination of cognitive abilities with crowdsourcing tasks. These prototypical microtasks (i.e., classification, sentiment analysis, counting, transcription) are representative of the tasks available in crowdsourcing marketplaces [45,46].

Classification is a category of microtasks that asks crowd workers to verify some characteristics according to a set of instructions or even the validation of certain content. As explained by Hettiachchi and colleagues [32], “In this task, crowd workers were presented with 16 paintings (primarily from The Metropolitan Museum of Art and the remaining from Flickr, all images licensed for public use) and were asked to identify and mark the items appearing in each painting from a given list of four items. Images represent different painting styles from different countries and contain one or more of the listed items.” Sentiment analysis was based on the crowd worker reading a sentence and analyzing the sentiment contained. A sentence’s sentiment was classified as either ‘negative’, ‘neutral’, or ‘positive’ in the version with personalization. In the version without personalization, it could be classified as ‘angry’, ‘sad’, ‘sarcasm’, ‘helpful’, ‘courage’, ‘happy’, or ‘patriotic’. Counting is a category of microtasks that asks crowd workers to count elements provided in an image, based on the description of characteristics with a set of instructions. As explained by Hettiachchi and colleagues [32]: “In this microtask, workers were presented with an image of a petri dish and asked to count malaria-infected blood cells. Workers were provided with specific instructions on how to differentiate an infected blood cell from an ordinary blood cell.” The accuracy of each counting microtask was determined by $\max(0,\;1\;-\frac{\left|response-groundtruth\right|}{\mathrm{groundtruth}})$. Transcription refers to the microtasks where the crowd workers typically analyzed an image with hand-written text and had to transcribe it. This used extracted images from The George Washington Papers at the Library of Congress [52]. The Levenshtein distance (*LD*) [53] was calculated between the response string and the ground truth, and accuracy was measured using $\max(0,\;1\;-\frac{2*LD}{\mathrm{length}(\mathrm{groundtruth})})$. The UI of the microtasks with personalization developed for this case study is presented in Figure 1 (the UI of the microtasks without personalization can be found here [32]).

Four experimental conditions were defined to answer the previously formulated research questions. This differed from prior works, and consisted of two main clearly distinguishable parts:Personalization of microtasks (with or without personalization). This condition focuses on the personalization of the interface. For this purpose, two interfaces were designed: one without any personalization and implying a normal difficulty, and the other one with personalization, where the content or the input elements were adapted to make it easier to solve each microtask. The elements required to make user interface (UI) adaptations were based on the ontology for cognitive personalization in a crowd work context proposed by Paulino and colleagues [38]. The cognitive knowledge that is represented in the ontology is based on mental functions, as defined by the international classification of functioning, disability and health (ICF), a framework for the classification of health and disability [1]. Furthermore, the ontology includes the concepts of microtasks, cognitive abilities, and types of adaptation, in order to personalize the interface to the crowd worker. To this end, an existing ontology called ACCESIBILITIC [54] was incorporated, which represents knowledge about accessibility and activity-centered design and includes a taxonomy with the classification of cognition-related concepts from the ICF scheme, which supported the personalization of the microtasks used in this case study. The group of microtasks with personalization (presented in Figure 1) was expected to have better microtask accuracy results when compared to the group without personalization (described in [32]).Cognitive profile (above or below the median overall score of cognitive abilities tests). The median value can be calculated based on the executive functions assessed using the administration of the cognitive ability tests. This value was used as a threshold for allocating the crowd workers to microtasks with or without personalization.

In general terms, this study intended to predict the performance when executing the different types of microtask. To this end, we defined four dependent variables related to the accuracy of microtasks (classification, counting, sentiment analysis, and transcription) and several independent variables (accuracy of Stroop, Flanker, Task Switching, Pointing and N-Back cognitive tests, response time, click details, and key presses of all cognitive tests and microtasks). The task fingerprinting was performed based on the click details and key presses, enabling the identification of behavioral traces. The development of the cognitive tests and the crowdsourcing microtasks was based on JSPsych [55], a JavaScript library that facilitates the development of online psychometric tests and which has been used in several research works in the domain of psychology [56,57,58]. The crowd workers were recruited from Amazon Mechanical Turk and a total of USD 1.2 was paid for an estimated time of 10 min to complete all cognitive tests and microtasks. This value followed the research trend of paying crowd workers according to the United States minimum wage [59].

After the results were obtained, a dataset was constructed from the performance in cognitive tests and microtasks, and then a deep regression neural network was developed. Deep learning (DL) and deep neural network (DNN) architectures have become the state-of-art methods in many fields of ML in recent years. DL has become one of the most researched and developed areas in recent years, with impressive results in every area it has been applied [60]. DL differs from other ML methods in the way it learns. DL automates much of the feature extraction. Thus, ML is more dependent on human intervention to learn and has a smaller number of layers. Consequently, it requires fewer data to learn patterns but does not achieve good results when compared with DL. ML utilizes AI techniques that teach the computer how to improve its behavior for a given task. Some techniques used for ML include support vector machines and decision trees [61]. The network consists of an input layer, multiple hidden layers, and an output layer. The nodes are fully connected, and the number of input layer nodes is equal to the number of features in the input data. Each hidden layer node is composed of neurons.

## 4. Results

For the study, we recruited 134 unique crowd workers (age: mean = 36.695, SD = 12.945; gender: male *n* = 81, female *n* = 53), split into two groups: microtasks with personalization (*n* = 69) and without personalization (*n* = 65). Although the UI was different for each microtask, the cognitive tests administered were equal in both groups, in order to obtain an accurate cognitive profile of the crowd worker. Task duration can be used as an indicator to identify non-competent crowd workers, as stated by Pei and colleagues [30]. However, this indicator is subjective in a remote environment, since it is difficult to determine if the crowd worker is taking time because he/she is carrying out the task or is doing another unrelated activity [26]. In this section, we describe indicators resulting from the interaction with the UI, which enabled the observation of how much time was actually allocated to the task and how much time was spent on pause.

### 4.1. Cognitive Tests

The average accuracy of all cognitive tests was similar in both the crowd workers who had microtasks with personalization (mean = 0.565, SD = 0.190) and those without personalization (mean = 0.556, SD = 0.1999). In fact, this was somewhat expected, since the same cognitive tests were administered to all of participants, without a noticeable influence from the type of microtasks performed. As expected, the same happened for the time duration (measured in seconds) of the cognitive tests (with personalization: mean = 1.119, SD = 0.467; without personalization: mean = 1.052, SD = 0.559), number of clicks (with personalization: mean = 22.943, SD = 9.987; without personalization: mean = 21.431, SD = 11.534), and number of key presses (with personalization: mean = 9.496, SD = 14.739; without personalization: mean = 12.240, SD = 22.013). Table 2 presents the results of the cognitive tests according to the four research conditions: personalization (batch of HITs with changes to the prototypical microtasks) and cognitive overall score (below or equal/above median of the average accuracy score extracted from the five cognitive tests).

Despite the understandable exception of some outliers, the cognitive tests generated similar distributions in the two different groups of crowd workers, which is to be expected, since both types of cognitive test had a considerable sample (for each group *n* ≥ 65; with personalization: *n* = 69, without personalization: *n* = 65). Although the results in both groups were expected to be similar for the cognitive tests, the same was not expected for the prototypical microtasks, due to the fact that the UI of the two versions of these microtasks was different, based on the personalization of the components of each HIT, as explained in the Methodology section.

### 4.2. Prototypical Microtasks

The difference between the versions with personalization (easy difficulty) and without personalization (normal difficulty) was based on the changes to the UI interface of the microtasks presented to the crowd workers. Figure 1 and Figure 2 present the box plots of the results obtained in the versions with personalization and without personalization, respectively. On the one hand, the personalized microtask UIs mainly featured multiple-choice inputs, making it easier for crowd workers to complete the tasks. On the other hand, non-personalized microtask UIs included open-ended answers, which could decrease the task success rate and increase the number of interactions with the interface. The primary variable observed was accuracy in determining the success of the crowd workers’ microtasks. Additionally, other metrics were examined for performing task fingerprinting based on the crowd workers’ behavior during task completion. Analyzing the average time taken to solve the microtasks was crucial for assessing the effort and interest of the crowd participants. Extremely short execution times may indicate disinterest or a task’s intuitiveness, while excessively long execution times usually signify a more complex execution and/or interpretation. Since these were multiple-choice tasks, it was expected that the average number of clicks would be around one.

Understandably, the accuracy of the microtasks without personalization (mean = 0.467; SD = 0.358) was lower than the accuracy of the counting microtasks in the pool of crowd workers who had personalization (mean = 0.546, SD = 0.369). Calculating the difference in the means of the accuracy, there was around a 10% decrease (while the standard deviation remained similar) on the version without personalization. Furthermore, we observed a similar pattern when considering classification microtasks with personalization (mean = 0.720, SD = 0.278) and without personalization (mean = 0.369; SD = 0.369). In both microtasks of counting and classification, it is important to emphasize that the content of the microtask (output shown to the crowd worker) was changed in the version with personalization, whereas the main image that the crowd workers were asked to count/classify in the version with personalization was cut in half compared to the original in the version without personalization, which eased the microtask difficulty. The accuracy of the transcription microtasks between the pool of crowd workers who had microtasks with personalization (mean = 0.892, SD = 0.211) and without personalization (mean = 0.445; SD = 0.348) showed an enormous difference. A possible reason for this difference was that not only the output of the microtask in the version with personalization was simplified, but also the input was changed from open-ended to multiple-choice. Moreover, the accuracy of the microtasks comprising sentiment analysis in the pool of crowd workers who had microtasks with personalization (mean = 0.431, SD = 0.300) showed an expressive difference when compared to the microtasks without personalization (mean = 0.303; SD = 0.305). In this microtask, the output remained the same and the only difference with the version with personalization was that the multiple-choice input only had three options, while the version without personalization had seven different options.

Looking at the results depicted in Figure 2 and Figure 3, we can observe the higher accuracy in the version with personalization when compared to microtasks without personalization. Subsequently, each version obtained a better accuracy for crowd workers with higher cognitive scores than those who had lower cognitive scores. One of the other metrics analyzed was the duration (measured in seconds) needed for the crowd worker to finish the microtask. Extrapolating to the microtasks involving counting (with personalization: mean = 10.087, SD = 6.380; without personalization: mean = 11.781, SD = 6.052) and sentiment analysis (with personalization: mean = 6.049, SD = 7.566; without personalization: mean = 6.455, SD = 6.250), we noted that the duration mean and standard deviation were similar between the versions. This did not happen in the microtasks comprising classification (with personalization: mean = 15.884, SD = 17.768; without personalization: mean = 30.351, SD = 21.395) and transcription (with personalization: mean = 9.694, SD = 8.522; without personalization: mean = 66.096, SD = 62.299), where the version with personalization demonstrated an much shorter time to accomplish the microtasks when compared with the original version without personalization. This mainly happened due to the differences in the complexity of the two versions of personalization, with expressive values in the case of transcription, where the input differed from open-ended to multiple-choice, with the former meaning that the crowd worker needed more time to insert their answer. As also shown in Figure 2, the time duration for the microtasks without personalization was lower in people with higher cognition, which translated into greater effectiveness.

In a broad sense, the number of clicks made on the mouse/touchpad can allow a better evaluation of the crowd worker’s interaction with the microtask interface. Similarly to what happened for the duration required to accomplish each microtask, both versions with and without personalization obtained an similar number of clicks in the microtasks involving counting (with personalization: mean = 4.710, SD = 1.783; without personalization: mean = 5.492, SD = 6.205) and sentiment analysis (with personalization: mean = 2.464, SD = 1.420; without personalization: mean = 2.523, SD = 1.288). The microtasks comprising classification (with personalization: mean = 3.275, SD = 3.072; without personalization: mean = 6.677, SD = 2.501) and transcription (with personalization: mean = 2.261, SD = 0.560; without personalization: mean = 6.507, SD = 6.792) showed an opposite trend. As also shown in Figure 3, the number of clicks in the microtasks without personalization was higher for crowd workers with a higher cognitive score compared the ones with a lower cognitive score, which also affected the accuracy of the microtasks performed.

The metric of key presses was similar to the click behavior, but instead the former corresponded to the interactions of the crowd worker with the keyboard. At a glance, the key presses registered similar results with both versions (with personalization: mean = 1.000, SD = 6.181; without personalization: mean = 1.600, SD = 4.860). However, the outcomes of the other types of microtask were different. For instance, the microtasks involving counting (with personalization: mean = 0.652, SD = 1.348; without personalization: mean = 4.231, SD = 10.811), classification (with personalization: mean = 1.319, SD = 4.230; without personalization: mean = 7.231, SD = 17.647), and transcription (with personalization: mean = 0.710, SD = 2.492; without personalization: mean = 79.415, SD = 70.321) obtained less key presses in the version with personalization when compared to the version without personalization. The outstanding result was for transcription, with a mean difference in the order of 100 times. This difference was explained by the change in the type of input for the two versions, from open-ended (version without personalization) to multiple-choice (version with personalization), with the former requiring the usage of a keyboard, while the latter only required the usage of the mouse/touchpad.

Table 3 presents the results of the identification of behavioral traces (task fingerprinting) comprised of average and standard deviation values, specifically based on the click data and the key presses. These behavioral traces were obtained based on how much time the crowd worker needed to perform a specific action. For this study, four different actions were specified: hurry actions (performed in less than 100 ms), confident actions (performed between 100 ms and 10,000 ms), hesitant actions (performed over 10,000 ms), and special actions (click or key presses in buttons/key presses that shortcut action, for example the key “alt” combined with other key). Regarding the behavioral traces identified from the click data, all four prototypical microtasks (i.e., counting, classification, transcription, and sentiment analysis) showed either more confident actions or less hurry actions in the version with personalization compared with the version without personalization. In the key press data, the number of almost every action (i.e., hurry, hesitant, confident and special) had increased in the version without personalization when compared with the version with personalization. It is necessary to take into consideration that the number of all key presses was greater in the former version, which indicates that the crowd workers had to perform more actions in that version, which highlights the increase of special actions performed in the version without personalization (e.g., in the transcription micro task it was helpful to use the CTRL + C and CNTRL + V combination to perform copy and paste).

### 4.3. Prediction Based on Microtask Fingerprinting with a Deep Learning Model

The results obtained from the cognitive tests and the prototypical microtasks (with or without personalization) provide us various insights regarding the optimization of microtask assignment. In addition, we introduced the usage of microtask fingerprinting through the identification of crowd workers’ behavior during the interaction with the UI, in order to complement the administration of the cognitive tests in a crowdsourcing setting. The validation of the latter relied on the application of a DL model based on the results obtained from the study (i.e., cognitive tests and prototypical microtasks with at least four main variables: accuracy, duration, click count, and key presses). We evaluated the performance of several deep learning (DL) models with a varyied number of layers in predicting microtask accuracy. Our results showed that the models with many layers were prone to overfitting, performing well on the training dataset but failing to generalize to unseen data. Meanwhile, models with fewer layers were unable to achieve satisfactory results, given the complexity of the input/output mapping. As we did not have a predetermined number of layers that was deemed appropriate, we employed a trial-and-error approach until we found a configuration of layers that yielded the optimal performance. In the end, the number of layers used in the model was determined by the need to balance accuracy and generalization, rather than using a pre-determined value.

As an example, one DL model used in our study had a sequential architecture with dense layers and ReLU and TanH activation functions. The model had 105 input dimensions and the layers with 256, 128, 64, 32, 16, and 8 neurons. We compiled the model using the mean squared error loss function and Adam optimizer. After a number of iterations, the DL model was evaluated on approximately 2000 rows of data from 134 unique crowd workers. Results were produced, so we used less average values and obtained more raw indicators (e.g., the crowd worker had to perform 5 ‘pointing’ cognitive trials and 3 ‘transcription’ microtasks; we only had one row of data from the total averages but if we unfolded the data from the 5 rows of cognitive trials using the 3 rows of microtasks performed, a total of 15 rows of data from a unique crowd worker could be obtained).

After the design and compilation, our dataset was split into 70% for training and 30% for testing, as we did not use a separate validation set, due to having a small dataset. We avoided overfitting by utilizing techniques such as early stopping and monitoring of training loss. Although we did not use a validation set in this study, cross-validation should be incorporated in future works, to further evaluate the performance of our models, in case the dataset does not grow. Although our dataset was small and the data complex, we considered our results to be positive. However, it is worth noting that the small size of our dataset may have impacted the generalizability of our model’s performance.

Our model was trained for 700 epochs, with a batch size of 20. The accuracy of our model was evaluated on the test dataset, achieving 95% when executing the model on the entire dataset instead of a single cognitive test with a single microtask (where some authors found a good relationship [32,33]). To calculate the accuracy, we generated predictions for the test data and then scaled them back to their original price scales. We used the absolute percent error to quantify the deviation between the predicted and actual values, with a perfect prediction being an exact match. The need to apply a holistic approach when dealing with this dataset was related to the focus of the crowd worker, i.e., when observing all cognitive tests, a crowd worker can be focused on each microtask and spend more time on it, as also corroborated by Paulino and colleagues [39].

Deep learning neural networks are likely to quickly overfit a training dataset with few examples, which occurred in our case. Ensembles of neural networks with different model configurations are known to reduce overfitting but require additional computational expense for training and maintaining multiple models. A single model can be used to simulate cases where many different network architectures are involved, by randomly dropping out nodes during training. This is called dropout and offers a computationally very cheap regularization method for reducing overfitting and improving generalization. We introduced dropout layers between every dense layer with a rate of 0.15, with a float number between 0 and 1 representing the fraction of the input units to drop. The activation functions used were TanH for the input layer and ReLu for the other layers. As Figure 4 depicts, the model was then compiled using the mean_squared_error as the loss function and Adam as the optimizer. The loss function is a measure of how well the prediction model performs in terms of being able to predict the expected value. The optimizer is an algorithm used to change the attributes of a neural network, such as the weights and learning rate, to reduce losses.

## 5. Discussion

### 5.1. RQ1: How Can the Administration of Cognitive Ability Tests, Specifically in the Evaluation of Executive Functions, Be Used to Allocate Crowd Workers to Microtasks with or without Personalization?

The study presented in this article was basically divided into two parts: the cognitive ability tests adapted for a crowdsourcing setting, and the prototypical microtasks whose design was presented to the crowd worker with or without personalization. Cognitive tests were distributed equally to the crowd workers who performed both versions of the prototypical microtasks. As expected, the cognitive tests had a similar distribution in both groups of crowd workers. Although the results in both groups were expected to be similar for the cognitive tests, the same was not expected to happen for the prototypical microtasks, due to the personalization level of the UI. Accuracy was the variable that served as a basis to determine the cognitive score and the success of the microtasks performed by the crowd workers. The order of higher to lower accuracy obtained in all microtasks was set as follows: the version with personalization and higher cognition score, version with personalization and lower cognition score, version without personalization and higher cognitive score, and version without personalization and lower cognitive score. This hierarchy of accuracy and cognitive score in the different versions could be used to match each crowd worker to the most suitable level of personalization. In this study, the version with personalization contained easier tasks. Nevertheless, from the requester’s perspective, the data quality could be more beneficial if the crowd worker was identified as someone who possessed the necessary abilities to perform harder tasks (in this situation, the version without personalization would be used). However, the crowd worker can be easily demotivated if they have a lot of trouble in performing the tasks without personalization [62]. The goal of optimizing this task assignment process is to match crowd workers to microtasks with a suitable difficulty level, so that they can perform the microtasks while staying motivated. As a result, the requester can obtain outcomes of crowdsourcing campaigns with the expected quality [33]. The usage of optimal task assignment appears as a feasible option in a crowdsourcing setting [63], and the administration of cognitive abilities tests can be applied to support the process of allocating crowd workers to microtasks, while taking into consideration the level of difficulty.

### 5.2. RQ1.1: Is a Deep Learning Model Able to Allocate Crowd Workers to Microtasks with Different Characteristics of UI and Complexities of the Content Presented?

Our findings underscore that a DL model can be built to predict whether it is necessary to personalize microtasks to guarantee a good task performance. In this model, the efficiency data of the microtasks were included, as well as the results obtained in the cognitive tests. Moreover, it was possible to obtain a prediction score of the microtask accuracy in the model with a 95% accuracy. We compared the results obtained with a study that carried out task assignment considering cognitive abilities. In particular, Hettiachchi et al. [32] built individual relationships between cognitive tests and microtasks. In the model we built, we applied a holistic approach and put all the cognitive tests together with each microtask, and we still managed to obtain a good prediction result. An explanation for this positive result may be that the focus of the crowd worker was more on microtasks and not so much on the cognitive tests, as they could have been be more comfortable doing the prototypical microtasks that are often found in the crowdsourcing marketplaces and thus need more time to accomplish them, as corroborated in [39].

### 5.3. RQ2: Can Task Fingerprinting, Based on the Identification of Behavioral Traces, Be Used to Allocate Crowd Workers to Microtasks with or without Personalization?

The prediction DL model included the data of microtask fingerprinting to complement the administration of the cognitive tests in a crowdsourcing setting. The DL model included the data of the cognitive tests and the prototypical microtasks with at least four main variables: accuracy, duration, click count, and key presses. The latter three were the general indicators of the crowd worker behavior, specifically when interacting with the UI to perform each microtask. The usage of this data on the DL model and the subsequent positive prediction result indicated a feasible research path of studying crowd worker behavior based on the interaction data logs that support an optimal microtask assignment. In microtasks comprising personalization, these metrics (duration, clicks, key presses) had similar patterns, whether considering crowd workers with low or high cognition, only obtaining a significant difference in effectiveness (with the exception of the sentiment analysis task). The time required for the crowd workers to carry out microtasks is an important factor for assessing their effort and interest, taking into account that an excessively short time may be associated with situations with a lack of interest on the part of the crowd worker, but also be related to tasks that are easy to understand. On the other hand, excessive durations are associated with more complicated tasks (whether in terms of execution or even interpretation of the task content). For example, the classification and transcription tasks with personalization required shorter execution times when compared to the original version without any personalization. This happened mainly due to the differences in the complexity of the two versions of personalization, where the input differed. Taking the transcription microtask as an example, the input was changed from open-ended to multiple-choice, with the former requiring that the crowd worker spent more time to choose their answer. Furthermore, as was expected in microtasks without personalization, the duration of performance in the microtasks was lower in people with higher cognition than with low cognition, which translated into a greater effectiveness.

In terms of the number of clicks made, the personalized microtasks predominantly featured multiple-choice answers, where the crowd worker could only select one option. Thus, it was expected that the number of clicks would be approximately one. However, some records displayed more than two clicks, potentially due to the latency of the crowd worker’s internet connection. Another plausible explanation could be that crowd workers clicked more than necessary, assuming that the system did not recognize their response. This emphasizes the significance of cross-referencing the obtained data, such as the duration of actions during the task and the number of clicks. This approach helps to better identify crowd workers’ behavior and comprehend the contextual factors impacting task performance. Furthermore, this data-crossing to better infer the behavior of crowd workers has been performed with success in other crowdsourcing studies [30]. Concerning the prototypical microtasks, we found that the classification and transcription tasks obtained polarized results in the versions with and without personalization. The number of clicks in microtasks without personalization was higher in crowd workers with a higher cognitive score, in contrast to the those with a lower cognitive score, which also affected the accuracy of the microtasks performed. The metric of key presses was similar to the click results, but instead the former corresponded to the interactions on the keyboard. The microtasks involving counting, classification, and transcription showed a lower number of key presses in the version with personalization. The most significant result was for transcription, with the difference in the average value being in the order of 100 times. This difference was explained (with a similar reason as regarding the duration of the microtask) by the change of the type of input in the two versions, from open-ended (version without personalization) to multiple-choice (version with personalization).

Responding to RQ2, it is possible to use task fingerprinting to allocate crowd workers to different levels of difficulty in microtasks. This was verified by the relationship obtained in this study between the identification of crowd workers’ behaviors (i.e., through metrics of interaction with the UI, such as duration, clicks, and key presses) and the accuracy of microtasks (i.e., measuring the success of the proposed tasks). These considerations confirm what has been described in the scientific literature (i.e., accuracy can only be predicted by identifying crowd workers’ behavior during the task) [26,30]. This can be particularly useful in crowdsourcing campaigns where there is no ground-truth value to assess the accuracy of the task at the outset, and where it is necessary to use other metrics (e.g., task fingerprinting) to assess the quality of responses [64].

## 6. Conclusions

Microtask fingerprinting in crowdsourcing presents an opportunity to identify crowd worker behaviors, which can be crucial for improving crowdsourcing campaign outcomes. This information can also serve as a parameter in task assignments, benefiting both the requester and crowd worker. While accuracy is the primary metric for evaluating crowdsourcing work quality, it may not always be feasible, particularly in open-ended campaigns with a broad range of responses and no ground-truth values. With the identification of behaviors, it is possible to predict whether crowd workers are diligent in completing different prototypical microtasks. Previous research used cognitive personalization within the scope of task assignment, first assessing cognitive capabilities and then mapping or adapting microtasks optimally to each crowd worker. The study presented in this article attempted to combine both approaches through investigating the possibility of complementing cognitive tests with task fingerprinting, in order to personalize microtasks. We deployed batches of cognitive tests (based on cognitive abilities, specifically executive functions) and prototypical microtasks in two HITs, which differed in meeting two conditions of the between-subjects study design: microtasks without personalization (normal difficulty) and with personalization (easy difficulty, where the inputs and/or the content was simplified). Both HITs used the same cognitive tests, and only differed in the microtask difficulties.

The results obtained in this study allow us to state that a DL model can be built to predict whether it is necessary to personalize microtasks to guarantee good performance. In this model, the microtask fingerprinting data of the microtasks were included, as well as the results obtained in the cognitive tests, and it was possible to obtain a prediction score of the microtasks accuracy in the model with a 95% accuracy. The usage of optimal task assignment appears to be a feasible option in a crowdsourcing setting, and the administration of cognitive ability tests can be applied to support the process of allocating crowd workers to microtasks with varying levels of complexity. Furthermore, it is possible to use task fingerprinting to allocate crowd workers to different levels of difficulty in microtasks. This was verified by the relationship obtained in this study between the identification of crowd worker behavior (i.e., through metrics of interaction with the UI, such as duration, clicks, and key presses performed) and the accuracy of microtasks (i.e., measuring the success of the proposed tasks). In future work, it is intended to validate the main findings of this study in an on-the-fly task assignment scenario. This will allow the real-time allocation of crowd workers based, not only on the cognitive score obtained from the cognitive tests, but also based on the crowd workers’ behaviors while they are performing the microtasks. While we achieved an accuracy of 95%, it is important to note that the small dataset size may have limited the model’s performance. In future work, we plan to address this limitation by collecting a larger dataset, to further evaluate the model’s performance and increase the confidence in the results. Additionally, this article could motivate other studies, to investigate if task fingerprinting could complement the administration of cognitive ability tests for accurately evaluating the executive functions of crowd workers. Furthermore, with the DL model developed, it would be interesting to assess the usage of task fingerprinting to help predict the executive functions of crowd workers, which would represent an innovative way of measuring cognitive abilities in a crowdsourcing setting.

## Figures and Tables

**Figure 1 sensors-23-03571-f001:**
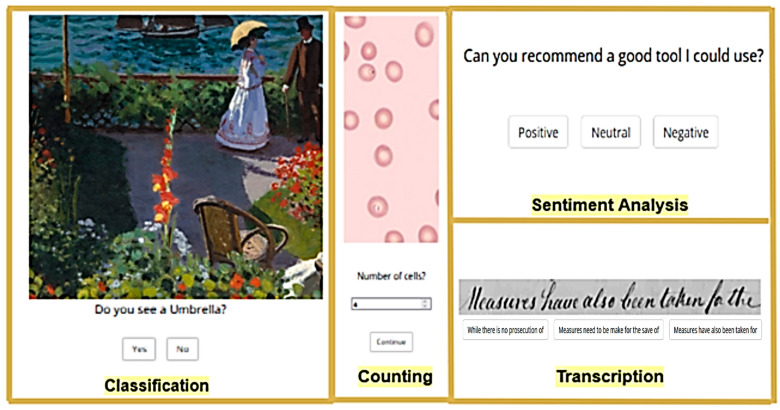
UI of microtasks (classification, counting, sentiment analysis, and transcription) with personalization developed for the case study.

**Figure 2 sensors-23-03571-f002:**
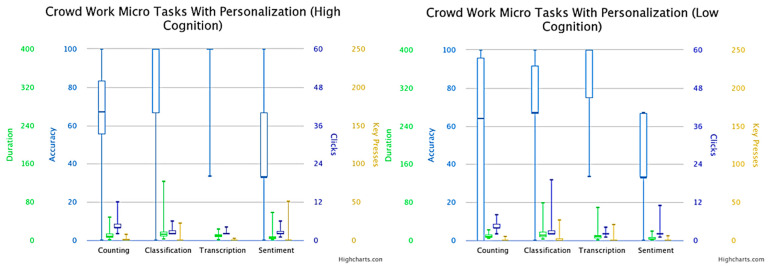
Box plot of microtask accuracy with personalization (easier difficulty) for crowd workers with a cognitive ability score equal or above median (**left**)/below median (**right**).

**Figure 3 sensors-23-03571-f003:**
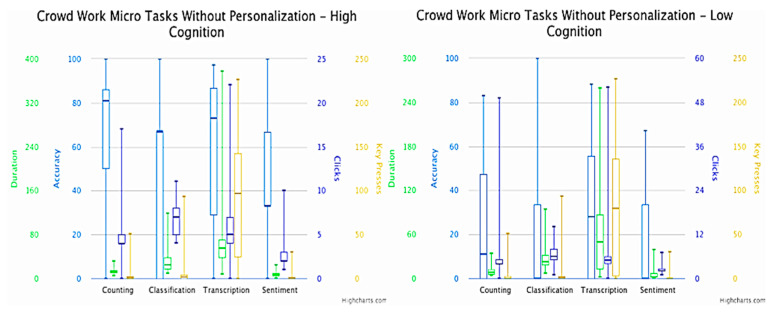
Box plot of microtask performance without personalization (normal difficulty) for crowd workers with a cognitive ability score equal or above median (**left**)/below median (**right**).

**Figure 4 sensors-23-03571-f004:**
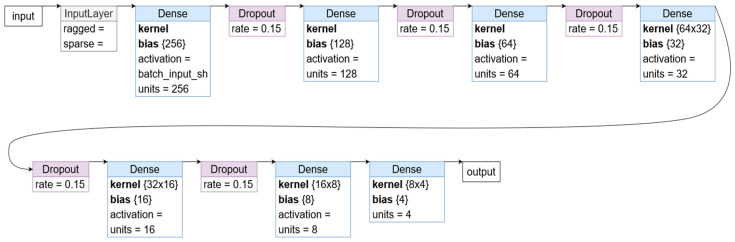
Visual representation of the developed DL model.

**Table 1 sensors-23-03571-t001:** Summary of the microtasks applied in this case study, consisting of cognitive tests and their positive relationship with the microtask (according to Hettiachchi and colleagues [33]).

Cognitive Test	Executive Function			Main Finding	Description of the Cognitive Test	Indicated Microtasks
	Cognitive Flexibility	Inhibition Control	Working Memory			
Flanker [47]		X		Assesses the ability to override the prepotent answer for incongruent elements.	Composed in trials of images showing five arrows. The congruent version has all arrows in the same direction. The incongruent version has the middle arrow pointing in the opposite direction.	Classification, Counting, Sentiment Analysis
N-Back [48]			X	Check if the individual can keep up with a sequence of stimuli	Presents a sequence of letters (each one represents a trial) and asks which one matches the three previous trials.	Classification, Counting
Pointing [49]			X	This test evaluates the ability to memorize a series of recent actions.	Composed of three to twelve squares, the individual must find each square, which may contain a visual reward, without repeating the previously clicked square.	Classification, Counting
Stroop [50]		X		Identical to the Flanker test, but in this version, it has words and different (or same) colors.	The words are presented with the name of a color. This test is composed of three diverse types of trial: congruent, incongruent, or unrelated.	Classification, Counting, Sentiment Analysis
Task-Switching [51]	X			Obligates the participant in some trials to change the question, by answering only if in some squares there is a vowel or a number even.	The letters and numbers are presented in four aligned squares.	Transcription

**Table 2 sensors-23-03571-t002:** Results of the cognitive tests, comprised of average and standard deviation values (both have 3 decimal digits) in the four research conditions of personalization and cognition (‘W/o’—without, ‘W/’—with, ‘pers’—personalization, ‘A.’—accuracy, ‘D.’—duration, ‘C.’—clicks, ‘K.’—key presses).

Research Condition	Flanker				N-Back				Pointing				Stroop				Task-Switching			
	A.	D.	C.	K.	A.	D.	C.	K.	A.	D.	C.	K.	A.	D.	C.	K.	A.	D.	C.	K.
**W/pers. and <** **median cognit.**	0.425 ± 0.260	1.262 ± 0.548	0.606 ± 2.633	12.545 ± 10.935	0.328 ± 0.180	0.968 ± 0.566	0.121 ± 0.415	9.000 ± 18.655	0.216 ± 0.170	0.779 ± 0.354	135.909 + 52.162	9.303 ± 23.115	0.855 ± 0.203	1.385 ± 0.298	0.121 + 0.415	7.909 ± 9.586	0.409 ± 0.139	1.178 ± 0.584	0.121 ± 0.415	16.455 ± 24.160
**W/pers. and ≥ median cognit.**	0.856 ± 0.183	0.945 ± 0.467	0.000 ± 0.000	10.333 ± 17.155	0.483 ± 0.180	1.172 ± 0.479	0.000 ± 0.000	7.056 ± 13.630	0.567 ± 0.315	0.792 ± 0.401	92.528 ± 43.663	3.417 ± 8.768	0.960 ± 0.098	1.129 ± 0.291	0.028 ± 0.167	6.472 ± 7.077	0.551 ± 0.176	1.584 ± 0.681	0.000 ± 0.00	12.472 ± 14.306
**W/o pers. and <** **median cognit.**	0.454 ± 0.285	1.109 ± 0.453	0.313 ± 1.768	23.938 ± 45.173	0.396 ± 0.201	0.686 ± 0.522	1.500 ± 6.730	13.375 ± 29.160	0.258 ± 0.224	0.920 ± 1.125	126.219 ± 71.174	27.031 ± 64.597	0.656 ± 0.296	1.239 ± 0.551	0.063 ± 0.246	7.938 ± 8.277	0.411 ± 0.128	1.024 ± 0.788	0.031 ± 0.177	14.875 ± 13.102
**W/o pers. and ≥** **median cognit.**	0.863 ± 0.165	0.978 ± 0.389	0.000 + 0.000	7.455 ± 10.019	0.461 ± 0.184	1.082 ± 0.466	0.182 ± 1.044	3.879 ± 5.042	0.596 + 0.261	0.745 ± 0.370	85.970 ± 34.023	9.000 ± 27.393	0.955 ± 0.068	1.178 ± 0.237	0.000 ± 0.000	6.182 ± 6.182	0.515 ± 0.187	1.560 ± 0.687	0.030 ± 0.174	8.727 ± 11.181

**Table 3 sensors-23-03571-t003:** Results of the identification of behavior traces (task fingerprinting) comprised of average and standard deviation values (both have 3 decimal digits) in the versions with and without personalization (‘W/o’—without, ‘W/’—with, ‘pers’—personalization, ‘Hu.’— hurry actions, ‘He.’— hesitant actions, ‘co.’—confident actions, “sp.”—special actions).

Micro Task and Behavior/Version	Counting				Classification				Transcription				Sentiment			
	Hu.	He.	Co.	Sp.	Hu.	He.	Co.	Sp.	Hu.	He.	Co.	Sp.	Hu.	He.	Co.	Sp.
**W/pers. (clicks)**	0.014 ± 0.120	0.797 ± 2.090	1.681 ± 2.110	0.000 ± 0.000	0.028 ± 0.168	0.956 ± 3.465	0.637 ± 1.042	0.000 ± 0.000	0.869 ± 0.339	0.057 ± 0.235	0.188 ± 0.393	0.000 ± 0.000	0.898 ± 0.304	0.115 ± 0.403	0.405 ± 0.845	0.000 ± 0.000
**W/pers. (key-** **presses)**	0.014 ± 0.120	0.376 ± 0.940	1.086 ± 2.605	0.231 ± 1.456	0.000 ± 0.000	0.101 ± 0.546	0.260 ± 1.024	0.086 ± 0.331	0.000 ± 0.000	0.463 ± 3.616	0.101 ± 0.425	0.275 ± 2.168	0.000 ± 0.000	0.000 ± 0.000	0.913 ± 3.716	0.362 ± 2.196
**W/o pers.** **(clicks)**	0.000 ± 0.000	0.985 ± 3.21	0.892 ± 0.886	0.000 ± 0.000	0.000 ± 0.000	3.246 ± 5.111	1.107 ± 1.985	0.000 ± 0.000	0.830 ± 0.377	1.400 ± 3.831	1.215 ± 1.231	0.000 ± 0.000	0.784 ± 0.414	0.153 ± 0.592	0.384 ± 0.896	0.000 ± 0.000
**W/o pers.** **(key** **presses)**	0.030 ± 0.174	0.800 ± 2.237	1.800 ± 3.700	0.323 ± 1.160	0.000 ± 0.000	1.769 ± 6.022	0.738 ± 4.176	0.353 ± 1.931	0.076 ± 0.268	51.938 ± 45.670	6.830 ± 8.1076	3.953 ± 5.421	0.000 ± 0.000	0.138 ± 0.788	0.892 ± 2.298	0.215 ± 1.038

## Data Availability

Not applicable.

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
