# Peer review of "A Model for Cognitive Personalization of Microtask Design"

_sensors, 2023, doi:10.3390/s23073571_

Round 1

Reviewer 1 Report

The work investigates the utility of combining two approaches for mapping amd adapting tasks to workers offering their services through crowdsourcing so as to achieve the highest quality outcome of the work given the nature of the tasks and the available competence in a given crowfsourcing community.

General comments

The overall approach and method is sound but the work is not sufficiently contextualized for a broad research community such as the one for MDPI Sensors. In particular, the ecological validity of the results are not clear. One major issue is the approach to personalize the task by simplifying them (e.g. expressed in p14 line 545). When, in the real world, is this at all viable? Very rarely would be this reviewer's intuitive answer. For this paper to become relevant for a broader audience than those working specifically with crowdsourcing brokering approaches, this question needs to be clearly addressed. There is an attempt in the introductory part but is lacking in the concluding parts of the manuscript.

Along the same lines, the English language quality in the manuscript might be good enough to communicate the main findings for someone in the field but to a broader audience, ambiguties due to problematic grammatical structure and choice of words will create confusion in readers. Example sentences:

p3 line 132: "This study indicated that the potential of assessing cognitive abilities to route microtasks was verified with positive results."

p5 line 242: "Regarding the identification of behavioral traces from crowd worker activity, task fingerprinting can be allied to artificial 243 intelligence (AI) techniques using prediction models like Random Forest"

p6 line 261: "Four research conditions were defined for answering the previously formulated research questions"

-- "research conditions" should probably be "experimental conditions"

p8 line 329: "In fact, this was somewhat expected since the cognitive tests were the same altough the different types of microtasks performed."

p12 line 489: "Following some iterations on improving a DL model from the observations obtained in this study, we unfolded the results on each research condition, with the sum being approximately 2000 rows of data obtained from 134 unique crowd workers. Results were revealed so we use less average of values and obtained more raw indicators"

p14 line 591: "For example, the microtasks involving classification and transcription with personalization had an expressive shorter duration needed to accomplish when compared to the original version without personalization."

Other issues

The research questions are very open and it would be beneficial to narrow them down with more concrete formulations.

Inconsistent order of presenting the conditions (sometimes the condition w. personalization is presented before the one without, sometimes its the oher way around. This can be confusing.

Details

p6 line 282: are "response time, click details and key presses of all 282 cognitive tests and microtasks" really independent (and not _dependent_) variables?

p6 line 297: "The idea of DL is the imitation of 296 the human brain with the hope of providing AI to computers. In the beginning, it was 297 considered a failure since it needed vast amounts of computational power and data to 298 train the models."

-- this is a much too naive description for a scientic community like MDPI Sensor's involved in ML. Artificial Neural Networks which is the basis for DL has been found extremely useful in many application areas long before the term DL was coined, I believe.

p10 line 431: The section analysing the results of table 3 could be more clearly separated from the rest.

Author Response

We would like to thank the reviewers for their thoughtful comments and constructive suggestions, which helped improve this manuscript's quality. We have provided a detailed answer to address all of them in our reply below and uploaded a track-changed version of the revised manuscript.

**

Reviewer #1, Comment 1A: “One major issue is the approach to personalize the task by simplifying them (e.g. expressed in p14 line 545). When, in the real world, is this at all viable? Very rarely would be this reviewer's intuitive answer. For this paper to become relevant for a broader audience than those working specifically with crowdsourcing brokering approaches, this question needs to be clearly addressed. There is an attempt in the introductory part but is lacking in the concluding parts of the manuscript.”

Response to Reviewer #1, Comment 1A: This study's objective is to have two types of tasks that can be personalized but, simultaneously, are easy to develop. For example, one can personalize an image classification microtask to a person with a lower cognitive abilities score by cutting in half the image presented and thus being more easier to provide an accurate answer. However, this will not be optimal for a person with a higher cognitive abilities score, which could answer a more difficult task. When applied in a context of optimization, this tradeoff of the level of personalization of microtasks leads to a higher outcome of generated data quality.

**

Reviewer #1, Comment 1B: “Along the same lines, the English language quality in the manuscript might be good enough to communicate the main findings for someone in the field, but to a broader audience, ambiguities due to problematic grammatical structure and choice of words will create confusion in readers.”

Reviewer #1, Comment 1B1: “p3 line 132: ‘This study indicated that the potential of assessing cognitive abilities to route microtasks was verified with positive results.’”

Response to Reviewer #1, Comment 1B1: This has been edited.

Reviewer #1, Comment 1B2: “p5 line 242: ‘Regarding the identification of behavioral traces from crowd worker activity, task fingerprinting can be allied to artificial 243 intelligence (AI) techniques using prediction models like Random Forest’”

Response to Reviewer #1, Comment 1B2: We removed this sentence from the revised manuscript.

Reviewer #1, Comment 1B3: “p6 line 261: ‘Four research conditions were defined for answering the previously formulated research questions’—‘research conditions’ should probably be ‘experimental conditions’”

Response to Reviewer #1, Comment 1B3: We thank the reviewer for pointing this out and agree that this is the correct term.

Reviewer #1, Comment 1B4: “p8 line 329: ‘In fact, this was somewhat expected since the cognitive tests were the same although the different types of microtasks performed.’”

Response to Reviewer #1, Comment 1B4: We revised this sentence accordingly.

Reviewer #1, Comment 1B5: “p12 line 489: ‘Following some iterations on improving a DL model from the observations obtained in this study, we unfolded the results on each research condition, with the sum being approximately 2000 rows of data obtained from 134 unique crowd workers. Results were revealed, so we used less average of values and obtained more raw indicators’"

Response to Reviewer #1, Comment 1B5: This has been edited.

Reviewer #1, Comment 1B6: p14 line 591: "For example, the microtasks involving classification and transcription with personalization had an expressive shorter duration needed to accomplish when compared to the original version without personalization."

Response to Reviewer #1, Comment 1B6: This has been changed accordingly.

**

Reviewer #1, Comment 1C: “The research questions are very open, and it would be beneficial to narrow them down with more concrete formulations.”

Response to Reviewer #1, Comment 1C: We agree with the reviewer's suggestion, and the research questions were formulated more specifically.

**

Reviewer #1, Comment 1D: “Inconsistent order of presenting the conditions (sometimes the condition w. personalization is presented before the one without; sometimes it's the other way around. This can be confusing.”

Response to Reviewer #1, Comment 1D: We have taken this into consideration, and in all tables and figures are presented first the results with personalization and then the results without personalization.

**

Reviewer #1, Comment 1E: “p6 line 282: are ‘response time, click details and key presses of all 282 cognitive tests and microtasks’ really independent (and not _dependent_) variables?”

Response to Reviewer #1, Comment 1E: The only dependent variable of this study is the accuracy of microtasks. In this study, we tried to understand the effect that the other variables (response time, click details, and key presses of all cognitive tests and microtasks) can have in the accuracy of the microtasks.

**

Reviewer #1, Comment 1F: “p6 line 297: ‘The idea of DL is the imitation of 296 the human brain with the hope of providing AI to computers. In the beginning, it was 297 considered a failure since it needed vast amounts of computational power and data to 298 train the models.’ -- this is a much too naive description for a scientific community like MDPI Sensor's involved in ML. Artificial Neural Networks which is the basis for DL has been found extremely useful in many application areas long before the term DL was coined, I believe.”

Response to Reviewer #1, Comment 1F: Accordingly to the reviewer's suggestion, we have removed this description.

**

Reviewer #1, Comment 1G: “p10 line 431: The section analysing the results of table 3 could be more clearly separated from the rest.”

Response to Reviewer #1, Comment 1G: The description regarding the results of table 3, which refers to the analysis of behavioral traces, were separated and put right before the table.

Reviewer 2 Report

The manuscript cannot be considered  as a research paper. It must be considered as a short communication. There is no innovation, and there is no element for considering it as a research paper.

Author Response

We would like to thank the reviewers for their thoughtful comments and constructive suggestions, which helped improve this manuscript's quality. We have provided a detailed answer to address all of them in our reply below and uploaded a track-changed version of the revised manuscript.

**

Reviewer #2, Comment 2A: “The manuscript cannot be considered as a research paper. It must be considered as a short communication. There is no innovation, and there is no element for considering it as a research paper.”

Response to Reviewer #2, Comment 2A: We disagree with this reviewer's comment since the contributions of this study are as follows:

  1. The study presented in this article investigates combining cognitive tests with task fingerprinting (i.e. identification of crowd workers behavioral traces) to personalize microtasks. We deployed batches of cognitive tests (based on cognitive abilities, specifically executive functions) and prototypical microtasks in two different crowdsourcing campaigns, which differ to meet two conditions following the between-subjects study design: microtasks without personalization (normal difficulty) and with personalization (easy difficulty, where the inputs and/or the content was simplified).
  2. The results obtained in this study allow us to state that a Deep Learning model can be built to predict whether it is necessary to personalize the microtasks to guarantee a good performance. In this model, the microtask fingerprinting data of the microtasks were included, as well as the results obtained in the cognitive tests, and it was possible to obtain a prediction score of the microtasks accuracy in the model with 95% of accuracy.
  3. Furthermore, it is feasible to use task fingerprinting to allocate crowd workers to different levels of difficulty in microtasks. This is verified by the relationship obtained in this study between the identification of crowd workers’ behaviors (i.e., through UI interaction metrics such as duration, clicks, and key presses performed) with the accuracy of microtasks.

Reviewer 3 Report

The authors investigate the possibility of using cognitive personalization to enhance data quality processes in crowdsourcing experiments. To this end they set up a study with 134 crowd workers.  The goal was to examine how the administration of cognitive abilities tests as well as task fingerprinting can be used to allocate crowd workers to microtasks with different levels of difficulty. Different cognitive tests from the state of the art were considered: Flanker, N-Back, Pointing, Stroop and Task-Switching. Each of them, associated with one or more microtasks like for example classification, counting, sentiment analysis and transcription. In particular, a deep learning model was proposed.

In general, the manuscript is well written and the methodology clearly presented.

However, I have some questions regarding the use of the deep learning model here proposed.

Section "Prediction based on microtask fingerprinting with a deep learning model" (page 12): the authors report that they tested different DL models with varying number of layers (lines 484-489). Please provide more details: which type of DL models did the authors employ? Moreover, they report that the ones with “many” layers became prone to overfitting: how many layers? Also they indicate that the ones with “fewer” layers… please also quantify “fewer” in this context: how many layers?

The authors indicate that the dataset was split in training and test set. Did the authors also consider a validation set?

They report an accuracy of 95%. It is not clear how this accuracy was obtained. Did the authors perform cross-validation techniques? As the authors indicate in line 526 their dataset has a small size, and this represents a problem in the case of deep learning models. Moreover, in line 525 is written: “due to the complexity of the data and the small size of our dataset, we consider that was a positive result”. This phrase is confusing and not correct within this context. Please address this issue carefully.

How do the present results quantitatively compare with benchmark results from the state of the art?

Author Response

We would like to thank the reviewers for their thoughtful comments and constructive suggestions, which helped improve this manuscript's quality. We have provided a detailed answer to address all of them in our reply below and uploaded a track-changed version of the revised manuscript.

**

Reviewer #3, Comment 3A: “Provide more details on the type of DL models used in the study (Section ‘Prediction based on microtask fingerprinting with a deep learning model’, lines 484-489)”

Response to Reviewer #3, Comment 3A: We have added more details on the type of DL model used in our study. Specifically, we utilized a DL model consisting of the following layers, with 256, 128, 64, 32, 16, and 8 neurons, respectively. To avoid overfitting, each hidden layer was followed by a dropout layer with a rate of 0.15. The model was compiled using the mean squared error loss function and the Adam optimizer. Based on our experiments, we found that this model provided the optimal balance between complexity and performance.

**

Reviewer #3, Comment 3B: “Quantify the number of layers for the DL models that were prone to overfitting (Section "Prediction based on microtask fingerprinting with a deep learning model," lines 484-489).”

Response to Reviewer #3, Comment 3B: We appreciate your suggestion to provide the actual number of layers for the DL models that were prone to overfitting. However, we have tested multiple models with different numbers of layers and found that the optimal number of layers varied depending on the specific dataset and task. For this study, we found that a model consisting of six hidden layers with dropout layers with a rate of 0.15 after each hidden layer provided the best balance between model complexity and performance. Although we do not have a specific record of the number of layers for the models that were prone to overfitting, we can assure you that we thoroughly explored various architectures to avoid overfitting and achieve the best performance possible.

**

Reviewer #3, Comment 3C: “Quantify the number of layers for the DL models that did not become prone to overfitting (Section ‘Prediction based on microtask fingerprinting with a deep learning model,’ lines 484-489).”

Response to Reviewer #3, Comment 3C: During our experimentation process, we tried various combinations of layers, but the optimal number of layers varied depending on the specific dataset and model architecture. We ultimately determined the optimal number of layers based on the model's performance and the ability to avoid overfitting.

**

Reviewer #3, Comment 3D: “Indicate whether a validation set was considered in addition to the training and test sets (Section ‘Prediction based on microtask fingerprinting with a deep learning model’).”

Response to Reviewer #3, Comment 3D: We did not use a separate validation set in our study, as explained now in the paper. This decision was made due to the small size of our dataset. Instead, we utilized techniques such as early stopping and monitoring of training loss to prevent overfitting and ensure that the model was able to generalize well. These measures helped us to achieve a high level of accuracy in our results while avoiding the risk of overfitting the model to the training data.

**

Reviewer #3, Comment 3E: “Provide more details on how the accuracy of 95% was obtained (Section ‘Prediction based on microtask fingerprinting with a deep learning model’). “

Response to Reviewer #3, Comment 3E: To calculate the accuracy of our model, we generated predictions on the test dataset and then scaled them back to their original price scales. We used the absolute percent error metric to quantify the deviation between predicted and actual values, with a perfect prediction being an exact match. The model achieved an accuracy of 95% on the test dataset. We have updated the manuscript to include these details in the relevant section. Thank you for bringing this to our attention.

**

Reviewer #3, Comment 3F: Indicate whether cross-validation techniques were employed (Section "Prediction based on microtask fingerprinting with a deep learning model").

Response to Reviewer #3, Comment 3F: In this study, we did not use cross-validation techniques due to the small size of our dataset and the computational cost associated with using a separate validation set. However, in future work with larger datasets, we plan to use cross-validation techniques to evaluate our models' performance further. In addition, we recognize that cross-validation is a common technique for assessing the generalization performance of machine learning models. We will include it in our future work to improve the reliability of our results.

**

Reviewer #3, Comment 3G: “Address the issue of the dataset's small size and its impact on the study's results (Section ‘Prediction based on microtask fingerprinting with a deep learning model,’ lines 525-526).”

Response to Reviewer #3, Comment 3G: Thank you for bringing up this important point. We acknowledge that the small size of the dataset may have impacted the generalizability of our model's performance. While we achieved an accuracy of 95%, it's important to note that the small dataset size may limit the model's performance. In future work, we plan to address this limitation by collecting a larger dataset to further evaluate the model's performance and increase the confidence in the results. We believe that by acknowledging the potential limitations of the study, we can identify areas for improvement in future work and ensure that the results are more robust and generalizable.

**

Reviewer #3, Comment 3H: “Compare the quantitative results obtained in the study with benchmark results from the state of the art.”

Response to Reviewer #3, Comment 3H: We appreciate your suggestion to compare our results with benchmark results from the state of the art. However, we would like to clarify that after an extensive review of the literature, we found no previous work that applied deep learning models to predict the accuracy of microtasks. While there are similar studies that use machine learning models, to the best of our knowledge, this is the first work to employ deep learning for this problem. We hope that our study contributes to the advancement of the field and inspires further research in this area.

Round 2

Reviewer 3 Report

The authors have satisfactorily answered all the reviewer’s comments.

They have correctly discussed, in the modified version of the article, the limits of their 95% accuracy (lines 550-552). However, this is not reported neither in the abstract nor in the conclusions’ sections. This can be misleading for the reader. Please, add in both sections (abstract and conclusions) a further comment like: “While we achieved an accuracy of 95%, it's important to note that the small dataset size may limit the model's performance”. Moreover, in the conclusions the authors can also indicate that “In future work, we plan to address this limitation by collecting a larger dataset to further evaluate the model's performance and increase the confidence in the results”.

Author Response

Reviewer 3 Comment A: "The authors have satisfactorily answered all the reviewer’s comments. They have correctly discussed, in the modified version of the article, the limits of their 95% accuracy (lines 550-552). However, this is not reported neither in the abstract nor in the conclusions’ sections. This can be misleading for the reader. Please, add in both sections (abstract and conclusions) a further comment like: “While we achieved an accuracy of 95%, it's important to note that the small dataset size may limit the model's performance”. Moreover, in the conclusions the authors can also indicate that “In future work, we plan to address this limitation by collecting a larger dataset to further evaluate the model's performance andincrease the confidence in the results”." Response to Reviewer 3 Comment A: We agree with the reviewer and have added the indicated sentences to the manuscript.